# USEFULNESS-DRIVEN LEARNING OF FORMAL MATHEMATICS

## ABSTRACT

Creating an AI that can truly "do" mathematics requires more than just solving isolated problems. It must mimic the creative, progressive nature of human mathematicians, who build upon previous work to generate new knowledge. A crucial part of this process is proposing theorems that serve as useful building blocks for proving more advanced theorems. In this paper, we introduce UseFor, a novel framework that formalizes this notion of usefulness and demonstrates how it can be used to train a usefulness-driven AI mathematician. UseFor determines a theorem's usefulness based on two criteria: its reusability in subsequent proofs and its contribution to increasing proof likelihood. We integrate UseFor into the self-play conjecturing-and-proving setting of Minimo ((Poesia et al., 2024)). That is, starting from only axioms, we iteratively train conjecturers to propose useful formal statements and provers that explicitly reuse them when generating formal proofs. We experimentally evaluate this usefulness-driven self-play approach across three mathematical domains: arithmetic, propositional logic, and group theory. Our evaluation considers two metrics: intrinsic usefulness, which measures how often our trained provers reuse theorems, and extrinsic usefulness, judged by a state-of-the-art large language model and external provers like SMT solvers. Our results demonstrate that our usefulness-trained model effectively generates a large number of intrinsically and extrinsically useful formal theorems. For instance, our approach outperforms the original Minimo by 2.9 times in extrinsic usefulness for arithmetic. Our work highlights the significant potential of integrating usefulness in AI-driven mathematical discovery.

## 1 INTRODUCTION

Mathematical reasoning has long stood as a frontier challenge for artificial intelligence (Newell & Simon, 1956). While large language models (LLMs) have achieved rapid progress in formal theorem proving (Yang et al., 2024), most approaches depend on extensive human-written corpora of proofs and conjectures (Yang et al., 2023; Ying et al., 2025). This dependence limits the domains in which they can operate and prevents them from advancing beyond existing human knowledge. By contrast, human mathematicians build knowledge by conjecturing new statements and proving them, gradually extending their theoretical landscape without external supervision.

A natural question, therefore, arises: can we replicate this process automatically? Specifically, can an artificial agent, starting only from axioms, learn via self-play between conjecturing and proving, bootstrapping its own knowledge and progressively discovering new mathematics? This paradigm (McAllester, 2020) would eliminate the need for human data, enable exploration of domains where no proofs exist, and produce a scalable source of synthetic theorems for training future provers. Recent work such as (Poesia et al., 2024) shows early signs of this vision: by iterating between conjecture generation and Monte Carlo tree search (MCTS)-guided proof search, Minimo gradually learns to prove increasingly more difficult statements from scratch. However, as we argue below, merely increasing difficulty as defined in Minimo is not enough to drive true theory building.

Minimo, like most self-play provers, evaluates conjectures purely by their difficulty—the negative log-probability of successful proofs under the current prover policy. This encourages the generation of challenging statements, but overlooks whether they actually help prove other results. In practice,

we find that conjectures promoted solely by difficulty are rarely reused in later proofs and offer little leverage for solving harder targets.

We are inspired here by Bengio & Malkin (2024)'s perspective that the value of a theorem lies in its connection to other theorems. They noted that "*a crucial component of the usefulness of a new proven theorem t (in the context of previous theorems $\mathcal{T}(S)$) is how efficiently $\mathcal{T}(S) \cup \{t\}$ compresses the set of all provable mathematical statements $\mathcal{M}$*". Tao (2007) makes a similar point, observing that the *strength* of a theorem is best judged by "*testing it against a class of questions and problems that the theorem is intended to assist with solving*". Both perspectives highlight that valuable conjectures should be judged not in isolation but advance the prover's qualitative capabilities and accelerate cumulative theory building.

This motivates our core idea: to advance automated theory building, we need a tractable metric that approximates this relational value of conjectures. We introduce such a metric, UseFor, which defines a conjecture as *useful* if it both appears in the proof of a downstream target and increases the prover's success likelihood (log-probability) on that target. This dual criterion excludes trivial tautologies with no proving power and narrowly phrased statements with little applicability in other proofs, yielding a practical proxy for Bengio & Malkin (2024)'s compression perspective.

We leverage UseFor to develop a usefulness-aware self-play loop that builds directly on Minimo (Poesia et al., 2024). After each round of conjecturing and proving, we use UseFor to identify proven conjectures with the most relational value for proving other conjectures. These useful theorems are made available to the proving model as lemmas and weighted more heavily in training, guiding future conjectures and proofs toward structures that accelerate cumulative theory building.

We evaluate our usefulness-driven self-play framework on three domains: arithmetic, propositional logic, and group theory. Our results show that our approach generates a significant number of theorem usages and more useful conjectures compared to Minimo based on LLM-as-a-judge. For instance, for arithmetics, our approach outperforms Minimo by 2.9 times in producing useful conjectures. These findings indicate that usefulness is a stronger intrinsic signal than difficulty for guiding conjecture generation, and that usefulness-aware self-play offers a scalable path toward data-free theory exploration.

**Main contributions.** Our paper makes the following contributions:

- We formalize the notion of theorem usefulness as a dual criterion of *usage* and *improvement*, and propose a tractable procedure for measuring it within self-play (Section 3.2.1).
- We introduce a usefulness-aware self-play loop that augments Minimo by selecting conjectures according to relational usefulness rather than difficulty (Section 3.2.2).
- We present stabilization techniques (triviality filtering and novelty bias) that keep the loop from collapsing into tautologies or memorized variants (Section 3.2.3).
- We provide empirical results across arithmetic, propositional logic, and group theory, demonstrating that usefulness-driven conjectures are more reusable and lead to higher prover success rates than difficulty-based baselines (Section 4).

## 2 RELATED WORK

Our work is primarily related to prior bodies of work on mathematical conjecturing, tactic discovery, and theory exploration. Our approach is distinguished by the fact that our model is trained in a tabula rasa fashion, without any pre-existing examples, and evaluated on the theory exploration task.

**Mathematical conjecturing.** Our work is most closely based on Minimo (Poesia et al., 2024), which proposes a theorem-proving model in the Peano (Poesia & Goodman, 2023) formal language that is trained through iterative conjecturing and proving from scratch. (Polu & Sutskever, 2020) also propose a model that is trained via self-play, while (Dong & Ma, 2025) demonstrate the ability of the iterative conjecturing-proving paradigm to enhance a pretrained theorem prover. However, these works only use conjecturing as a means to improve the proof-search capabilities of the model, and do not attempt to evaluate the conjecturing abilities of the model directly. LeanConjecturer (Onda et al., 2025) proposes a model specifically designed for the conjecturing task, but uses a

pretrained LLM; in doing so, the ability of the LeanConjecturer model to generate novel conjectures cannot be faithfully evaluated due to inevitable contamination from pre-training data. Compared to these works, our approach evaluates conjecturing as a stand-alone task, while our tabula rasa setting allows us to definitively confirm the novelty of conjectures generated by our model.

**Tactic and premise discovery.** There is also a body of work concerning the task of tactic discovery, which aims to construct tactics in an interactive theorem prover setting that simplify proofs or otherwise enhance proving capabilities. TacMiner (Xin et al., 2025) proposes a method to find tactic simplifications in RCoq, given an existing high-quality corpus of proofs. Lego-Prover (Wang et al., 2023) and Seed-Prover (Chen et al., 2025) use already proven lemmas as a way to strengthen a theorem proving model, in the Isabelle and Lean 4 settings, respectively. However, all of these approaches require a dataset of high-quality, human-generated proofs, while our approach generates useful premises from scratch.

**Theory exploration using machine learning.** Finally, a third body of work is theory exploration using ML methods, the task of formulating interesting conjectures about a given problem domain (Johansson & Smallbone, 2021). We consider this problem to be the one our work addresses most closely. While a number of classical and neural approaches have been proposed for this task, existing neural methods work by training or finetuning a model based on an existing proof corpus (Urban & Jakubův, 2020). Lemmanaid (Alhessi et al., 2025) uses neuro-symbolic methods by finetuning a model with a subset of an existing proof library, and then evaluating it on another subset of conjectures. In search of a purely intrinsic approach in order to discover how a model could discover this usefulness without relying on human data, we distinguish ourselves by not training on external data, an approach similar to what has been done for SMT solvers (Gauthier & Urban, 2025).

## 3 Methodology

### 3.1 Base Self-Play Framework (Minimo)

A central challenge in building autonomous theorem-proving agents is the lack of human-labeled data. Unlike natural language or code, formal mathematics has limited corpora, and many target domains have essentially no prior datasets. There are two main approaches to address this bottleneck. One line of work leverages autoformalization, which translates large volumes of informal mathematics into formal statements, an approach that has already shown promise in practice. A more first-principles alternative is to remove reliance on pre-existing data entirely by using self-play: coupling[1] a conjecturer, which proposes candidate statements, with a prover, which attempts to establish them. Through repeated interaction, both components improve jointly in a closed loop, enabling progress even in domains with no human supervision or existing corpus.

Minimo (Poesia et al., 2024), implemented in the Peano environment (Poesia & Goodman, 2023), instantiates this idea. Starting from axioms alone, it alternates between conjecture generation and proof search. Over time, the prover strengthens by training on successful proof traces, while the conjecturer adapts toward statements near the boundary of provability. This process yields an automatically generated curriculum of increasing difficulty, with no reliance on human annotations. We summarize its core components next, as they provide the foundation on which our method builds.

### 3.1.1 Conjecturing

The conjecturer $\mathcal{C}_\theta$, with $\theta$ denoting the model parameters, generates statements in the Peano language, a dependently typed tree-based formal system with a finite action space (Poesia & Goodman, 2023). Each formula is represented as a well-formed term tree. To prevent invalid formulas, Minimo employs constrained decoding Poesia et al. (2021): at each step, candidate tokens are filtered so that only those extending the current tree into a valid continuation remain. This guarantees both syntactic and semantic validity, and prevents wasting prover efforts on malformed statements.

---

[1]In practice, coupling means using a single underlying model for both conjecturer and prover roles, as proposed by Poesia et al. (2024).

### 3.1.2 PROOF SEARCH

The prover $\mathcal{P}_\theta$ attempts to establish each conjecture using Monte Carlo tree search (MCTS), which is well suited to the combinatorial branching structure of formal proofs. A proof state $s$ encodes the current context, including the set of assumptions, the active subgoal, and any partially completed steps. At each state, the Peano environment defines the finite set of admissible inference actions. Guided by the prover's policy $\pi_\theta(a \mid s)$ and value estimates, MCTS expands trajectories

$$\tau = (s_0, a_0, s_1, a_1, \ldots, s_T)$$

starting from the initial state $s_0$. A completed trajectory corresponds to a valid proof of the conjecture. Its score is the log-likelihood under the prover's policy,

$$\ell(c) \;=\; \log p_\theta(\tau \mid c) \;=\; \sum_{t=0}^{T-1} \log \pi_\theta(a_t \mid s_t).$$

Less negative values of $\ell(c)$ indicate that the proof was expected under the current policy, while more negative values correspond to surprising but ultimately valid proofs. To exploit partial progress, Minimo applies *hindsight relabeling*: even when a conjecture cannot be proved in full, explored search trees are decomposed into valid subtraces corresponding to intermediate lemmas, which are then incorporated as additional training data (Poesia et al., 2024). This enlarges the training set and recycles computation that would otherwise be wasted on failed proofs.

### 3.1.3 CONJECTURING–PROVING SELF-PLAY LOOP

The conjecturer $\mathcal{C}_{\theta_i}$ and prover $\mathcal{P}_{\theta_i}$ interact in an iterative loop. At iteration $i$, the conjecturer samples a batch of $N$ candidate statements

$$\mathcal{Q}_i = \{c_1, \ldots, c_N\} \sim \mathcal{C}_{\theta_i}(\cdot \mid \mathcal{T}_i),$$

where $\mathcal{T}_i$ is the current theory consisting of axioms and previously promoted lemmas. For each $c \in \mathcal{Q}_i$, the prover attempts to establish it via MCTS:

$$(\text{proof}(c), \ell(c), \text{trace}(c)) \leftarrow \text{MCTS\_PROVE}(c; \mathcal{T}_i, \mathcal{P}_{\theta_i}),$$

where $\text{proof}(c)$ is a complete proof trajectory $\tau$ if one is found (or $\varnothing$ otherwise), $\text{trace}(c)$ is the explored search tree, and $\ell(c)$ is the log-likelihood of the trajectory under the prover's policy.

Conjectures are then stratified by empirical difficulty. Let $\mathcal{S}_i = \{c \in \mathcal{Q}_i : \text{proof}(c) \neq \varnothing\}$ be the set of successful conjectures, and let $q_{20}$ and $q_{50}$ denote the 20th and 50th percentiles of $\{\ell(c) : c \in \mathcal{S}_i\}$. Labels are assigned to each conjecture $c$ as

$$\text{label}(c) = \begin{cases} \text{``fail''}, & \text{if proof}(c) = \varnothing, \\ \text{``hard''}, & \text{if } \ell(c) < q_{20}, \\ \text{``easy''}, & q_{20} \leq \ell(c) < q_{50}, \\ \text{``trivial''}, & \ell(c) \geq q_{50}. \end{cases}$$

The dataset for iteration $i$ is then

$$\mathcal{E}_i = \{(\text{trace}(c), \text{label}(c)) : c \in \mathcal{Q}_i\},$$

which aggregates conjectures, proofs when available, and hindsight-relabeled subproofs extracted from failed searches. Both $\mathcal{C}_\theta$ and $\mathcal{P}_\theta$ are updated on $\mathcal{E}_i$, creating a feedback loop: the conjecturer shifts toward generating statements just beyond the prover's current reach, while the prover expands its competence from the resulting proofs. This difficulty-driven loop is the foundation upon which we build in Section 3.2, where difficulty is replaced with a more relational signal of usefulness.

### 3.2 USEFULNESS-AWARE SELF-PLAY LOOP

The self-play framework of Minimo provides a compelling basis for data-free theory exploration: starting from axioms, conjecturing and proving improve together in a bootstrapping loop. However, its training signal is limited to conjectural difficulty, measured as the negative log-probability of a

proof under the current prover. While effective for generating a curriculum of harder statements, this signal is ultimately syntactic. It rewards conjectures that are improbable under the model's local policy, but does not account for whether they connect meaningfully to other theorems explored so far. As a result, the system often promotes conjectures that are labeled as "hard" but not necessarily useful statements: isolated identities that stretch the prover temporarily but are rarely reused and add little structure to the theory. The log-probability score treats difficulty as an end in itself, overlooking the relational role that lemmas play in enabling further proofs and sustaining theory growth.

To address this limitation, we introduce a usefulness-based self-play loop. Instead of ranking conjectures solely by syntactic hardness, we ask whether incorporating a new lemma changes the prover's future behavior, specifically, whether it makes other statements easier to prove. Conjectures that are both provable and demonstrably beneficial in downstream proofs are promoted into the growing library, and their traces are used to train both the conjecturer and the prover. This shifts the learning objective from accumulating difficult but isolated statements to building a network of reusable ones, better aligned with the cumulative nature of mathematical discovery.

### 3.2.1 DEFINITION OF THE USEFULNESS METRIC

The perspectives of Bengio & Malkin (2024) and Tao (2007) converge on the idea that the value of a theorem is relational: it derives its significance not from truth alone, but from its effect on subsequent reasoning. Yet they articulate this in complementary registers. Bengio & Malkin (2024) frames usefulness in information-theoretic terms, proposing that a theorem acts as a *compression primitive*—its addition to a base theory reduces the description length of other proofs. Tao (2007) instead emphasizes the pragmatic dimension: the *strength* of a theorem is revealed only by confronting new problems and observing the range of arguments it simplifies.

While these views are philosophically aligned, neither directly yields a metric implementable within a self-play loop. Compression, though elegant, requires comparing description lengths over the unbounded space $\mathcal{M}$ of all provable statements, which is an intractable quantity in practice. Tao (2007)'s criterion, by contrast, presupposes a human mathematician's judgment in selecting "a class of questions and problems" against which to test strength. What is missing is a procedure that preserves the spirit of both notions while remaining computable for a prover–conjecturer system.

Our contribution is to bridge this gap by constructing an operational proxy for usefulness that can be applied iteratively inside the self-play loop. At a high level, the metric estimates a conjecture's capacity to expand the prover's effective reach: conjectures are useful insofar as their availability systematically reduces the effort of proving a benchmark set of targets.

Formally, let $\mathcal{B}$ be a benchmark set consisting of theorems that are difficult, but not impossible, for the prover to prove. In Section 3.2.2, we detail how we instantiate $\mathcal{B}$ using theorems generated internally by our self-play training loop. For each $b \in \mathcal{B}$, let $p_\theta(\tau_b \mid b)$ denote the prover's probability of producing a proof trajectory $\tau_b$ under theory $\mathcal{T}$, and let $p'_\theta(\tau_b \mid b)$ denote the same quantity when a candidate lemma $\ell$ is available. We say that $\ell$ is *useful* if there exists $b \in \mathcal{B}$ such that

(i) $\ell$ is invoked in the proof trace of $b$, and (ii) $\log p'_\theta(\tau_b \mid b) - \log p_\theta(\tau_b \mid b) > 0$.

Both conditions are essential: usage without improvement admits trivial tautologies such as $\forall x.\, x = x$, which the prover may frequently attempt but which yield no real progress. Only when the two conditions coincide do we identify lemmas that are genuinely structural.

To evaluate this criterion efficiently, we do not re-prove $\mathcal{B}$ for every lemma in isolation. Instead, given a set of newly proved conjectures $\mathcal{C}$, we subsample a subset of size $\lceil \sqrt{|\mathcal{C}|} \rceil$ and temporarily add them to the context. Each $b \in \mathcal{B}$ is then re-proved once under this extended theory. If a candidate $\ell$ appears in the proof of $b$ and the resulting log-likelihood improves relative to baseline, the gain is attributed to $\ell$. The aggregate score

$$U(\ell) \;=\; \sum_{b \in \mathcal{B}} \mathbf{1}\{\ell \in \mathrm{proof}(b)\} \cdot \max\{0, \log p'_\theta(\tau_b \mid b) - \log p_\theta(\tau_b \mid b)\}$$

is then used to rank candidates. Only the top $\rho$ fraction are promoted to the library, together with their associated proofs and hindsight traces. This provides a tractable mechanism for selecting conjectures that repeatedly demonstrate both reuse and measurable downstream gains.

**Illustrative scenario.** Consider arithmetic with multiplication defined inductively. Early in training, the prover may not yet know lemmas such as $x \times (y+1) = xy+x$. Without this fact, even simple targets like $(x+1) \times (y+1) = xy + x + y + 1$ require long derivations by repeatedly unfolding the definition of multiplication. Once $x \times (y+1) = xy+x$ is conjectured and proved, however, it can be applied directly, and many small multiplication–addition identities shorten dramatically. Our metric marks this lemma as useful precisely because it is both *used* in subsequent proofs and its presence *improves* the prover's success probability. Later discoveries, such as distributivity, compound this effect across broader families, but it is these intermediate stepping-stone lemmas that first enable steady cumulative progress.

### 3.2.2 TRAINING LOOP WITH THE USEFULNESS METRIC

We now describe how the usefulness metric is integrated into the conjecturing–proving loop. The outer structure mirrors Minimo (Poesia et al., 2024): in each iteration the agent generates conjectures, the prover attempts proofs via MCTS, and traces are collected. The crucial difference lies in how conjectures are filtered, promoted, and fed back into training. Whereas Minimo labels conjectures by proof log-probability percentiles and emphasizes those deemed "hard", our framework evaluates conjectures by their relational usefulness.

At iteration $i$, the conjecturer first proposes a batch $\mathcal{C}_i$, which is passed through a triviality filter to remove vacuous identities. Each conjecture is then attempted under the current theory $\mathcal{T}_i$ using MCTS, producing proofs, log-likelihoods, and hindsight examples. Following Minimo, conjectures are provisionally bucketed into "hard", "easy", and "trivial" categories by percentile of log-likelihood. Non-failing conjectures are collected as candidate lemmas.

The key departure comes in how the "hard" subset is treated. Rather than promoting the "hard" conjectures indiscriminately, we apply the *usefulness test* with them as the benchmark set $\mathcal{B}$, and with the set of all previously proven theorems $\mathcal{H}_i$ as our set of "potentially useful" lemmas. A random subsample $L_i \subseteq \mathcal{H}_i$ of size $\lceil \sqrt{|\mathcal{H}_i|} \rceil$ is drawn, and each benchmark $b \in \mathcal{B}$ is re-proven under the augmented theory $\mathcal{T}_i \cup L_i$. If a lemma $\lambda \in L_i$ is invoked in the augmented proof of $b$ and improves its log-likelihood relative to the baseline, the gain is added to its cumulative usefulness score $U_i(\lambda)$. Candidates are then ranked by $U_i(\lambda)$, and only the top $\rho$ fraction are maintained in $\mathcal{H}_{i+1}$ for future usefulness evaluations . Because $L_i$ is resampled at every iteration, different subsets of candidates are tested over time, so all conjectures eventually receive usefulness credit.

Finally, we assemble the training dataset $\mathcal{E}_i$. It includes the conjectures along with their percentile labels, their proofs, and the hindsight traces from the base loop as in Minimo. Additionally, we incorporate the useful lemmas and re-proving trajectories produced during usefulness testing. Specifically, each lemma deemed useful is added a conjecture under a "useful" category. We also include proof-search attempts that happened re-proving, which may involve lemma reuses. These additions help both the conjecturer and prover internalize the notion of "usefulness". The agent is updated on $\mathcal{E}_i$, and the promoted and untested lemmas are added to $\mathcal{H}_{i+1}$ for future iterations.

In summary, Minimo's curriculum is driven by proof difficulty under the current prover, whereas our loop is driven by demonstrable downstream impact. Only conjectures that are both *used* and *improve* benchmark proofs are promoted, producing a library that is not just deeper but more interconnected, with lemmas reappearing across proofs and compounding overall success.

### 3.2.3 OTHER IMPROVEMENT TECHNIQUES

Although the usefulness loop provides a stronger supervisory signal than difficulty alone, we observed recurrent failure modes in practice. In particular, the conjecturer may become trapped in local minima, repeatedly generating trivial identities or minor variants of existing statements. This behavior resembles exploiting shortcuts rather than genuine advancements. To enhance robustness and better align training with the intended objectives, we incorporate two additional techniques.

**Triviality filtering.** Before proof attempts, we remove conjectures that match heuristic patterns such as tautologies ($x = x$) or constant-only identities ($0 + 1 = 1$). Such statements can be discharged immediately without reasoning, yet they satisfy the usage criterion and would otherwise dominate the usefulness signal. Filtering them prevents the prover cycles from being wasted and prevents the model from collapsing toward vacuous but spuriously rewarding lemmas.

**Novelty bias.** To encourage structural diversity, we penalize the conjecturer for generating statements that share long prefixes with previously generated conjectures. This discourages local memorization and pushes exploration toward unexplored syntactic regions, thereby increasing the likelihood of uncovering genuinely new lemmas that expand the theorem library.

These techniques do not alter the usefulness metric itself, but regularize the conjecturer's proposal distribution. By filtering trivialities and discouraging near-duplicates, the system avoids spurious short-term rewards and maintains pressure toward conjectures that are both novel and reusable. Empirically, they improve the stability of the usefulness-aware loop, allowing the training distribution to shift steadily toward conjectures that promote cumulative theory building.

## 4 EXPERIMENTAL EVALUATION

We now present our experimental evaluation. Our goal is to assess whether UseFor demonstrates the essential qualities of a desirable reasoning system: (a) the ability to accumulate knowledge within the self-play training loop, (b) the ability to generate conjectures that are useful both internally (for self play) and externally (for the outside world), and (c) whether our usefulness-driven training is necessary. In more detail, we would like to address the following research questions:

- **RQ1**: *Can the prover reuse theorems proven in previous iterations to prove current conjectures?* Reuse is essential for cumulative theory building: without it, a system risks repeatedly rediscovering tautologies or isolated results, rather than developing an interconnected body of theory.

- **RQ2**: *Do likelihoods of theorem-reusing proofs increase across multiple iterations?* This would signify that during the training process, the prover is gradually gaining more capabilities and confidence in theorem reuse.

- **RQ3**: *Are the conjectures useful beyond self-play?* Extrinsic usefulness tests whether the system discovers theorems a mathematician would value, rather than artifacts of the training loop.

- **RQ4**: *Is the usefulness metric essential for conjecturing quality?* Without it, does the model discover interesting theorems? This matters because the entire training loop relies on this metric as its guiding signal.

### 4.1 EXPERIMENTAL SETUP

**Evaluation metrics.** In light of the above interesting research questions, we employ two complementary metrics designed to capture structural usefulness:

- *Intrinsic usefulness*: measured as the number of times a previously proven theorem is reused during usefulness testing. A high score indicates that the system is both conjecturing and successfully reusing theorems in its own proving process.

- *Extrinsic usefulness*: measured via an LLM-as-judge (GPT-4.1), which rates conjectures for mathematical value after a deduplication step that removes near-duplicates (details can be found in Appendix B). We also require that the conjectures can be proved by an external automated prover based on the Z3 STM solver (De Moura & Bjørner, 2008), which is effective on our self-play-generated conjectures. This metric evaluates whether conjectures would be judged useful by a human mathematician, beyond the system's internal dynamics.

Poesia et al. (2024) introduced intrinsic metrics based on the proof difficulty of internally generated conjectures, and extrinsic metrics based on the prover's success rate on human-written theorems. However, their metrics are unsuitable for our goals. First, they primarily measure proving performance, whereas our focus is on conjecturing. In addition, they evaluate isolated statements, while our metrics capture how the conjectured statements interact and build on each other.

**Baselines.** We compare UseFor against two baselines:

- "Base Minimo": The original Minimo algorithm (Poesia et al., 2024) without any modification, enabling a direct comparison with prior work.

- "No usefulness training": Our full approach but without the usefulness training described in Section 3.2.2, while retaining the improvements in Section 3.2.3. This isolates the contribution of usefulness training.

**Mathematical domains.** We conduct our evaluation on three mathematical domains: (i) arithmetic, (ii) propositional logic, and (iii) group theory. This setup follows Poesia et al. (2024), enabling clear comparison. The axioms of these domain are directly taken from Poesia et al. (2024) and are presented in Appendix A. All models in our experiments were fully bootstrapped from these axioms in Peano (Poesia & Goodman, 2023) without relying on any other external data.

**Model and self-play configurations.** We follow the setup of Minimo Poesia et al. (2024) and use an 8.45M-parameter GPT-2 model for both conjecturing and proving. All models are trained starting from scratch, ensuring that any generated theorem are genuine "discoveries". We run training for 10 iterations (compared to 5 in Minimo), as our approach benefits from cumulative improvements across iterations. In each iteration, we generate 200 conjectures. We perform proof search using MCTS with a budget of 1000 expansions per conjecture. All experiments are repeated three times, and we report averaged results to account for stochastic variations. Additional training details are provided in Appendix C.

## 4.2 EXPERIMENTAL RESULTS

**(RQ1) The model reuses previously proven conjectures.** Reuse is a key indicator of cumulative reasoning: a system that fails to apply previously proven theorems risks stagnating in isolated rediscoveries, rather than developing an interconnected theory. In our experiments, UseFor shows a steady increase in lemma usage during usefulness testing (Figure 1). Although the first few iterations provide little signal, usage accelerates in later iterations, demonstrating that the model progressively conjectures more useful theorems and becomes increasingly capable of applying them. This trend is consistent across all domains, and we expect it to persist with additional iterations. Since Minimo does not use previously proven theorems, its intrinsic metric is identically zero; we therefore do not include it as a baseline here.

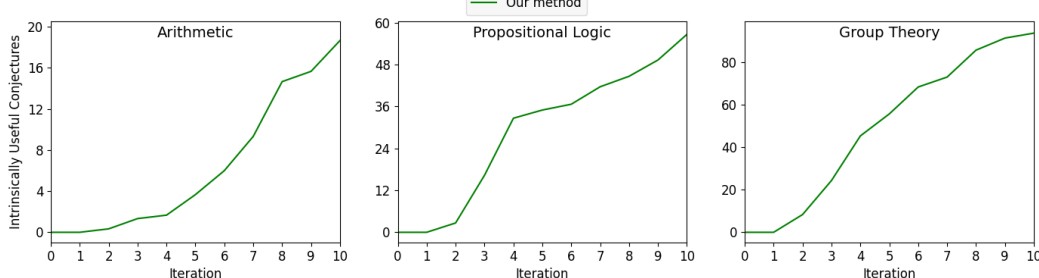

Figure 1: Intrinsic Evaluation: Total theorem use count with increasing iterations.

**(RQ2) The model grows increasingly more confident in theorem reuse.** As training progresses, our model grows increasingly confident in its use of previously conjectured lemmas, as evaluated by the average log-probability of proofs where at least one previously conjecture was used (Figure 2). This aligns with the significant increase in intrinsically useful conjectures across multiple iterations, as shown in Figure 1, and shows that the UseFor training objective is effective in encouraging the model to use previously proven lemmas. As we notice an upwards trend as iterations continue, this also demonstrates that our lemmas become more difficult to prove as time goes on, as earlier provers assign low probabilities to them. In addition, our

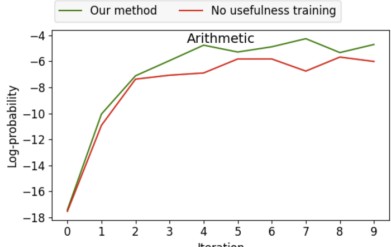

Figure 2: Average log-probabilities of proofs where a previous conjecture was used, across prover iterations.

model also achieves consistently higher probablities than "No usefulness training". This shows that training on usefulness testing improves the prover's confidence in its lemma reuse proofs.

**(RQ3) The conjectures are extrinsically useful**  In early iterations, UseFor quickly identifies many "easy" theorems accessible through shallow search. Crucially, usefulness continues to increase in later iterations, indicating that the system discovers progressively deeper and less trivial results. The growth of this metric empirically suggests that UseFor can generate theorems that are regarded as useful in the real-world by LLM-as-judge (and thus human). In Appendix B.5, we provide examples of extrinsically useful theorems conjectured by our model. In the case of group theory, our model conjectures less extrinsically useful theorems than base Minimo (average across 3 runs). However, using an LLM in order to compare the union of all conjectures generated during our 3 runs (detailed in Appendix B.3), we are able to find that in total, our method generates 23 conjectures not semantically equivalent to those of Minimo. Meanwhile, Minimo generates 21 conjectures not equivalent to ours and 64 conjectures are generated by both models. Therefore, our model still remains diverse and provides complementary results to those of Minimo.

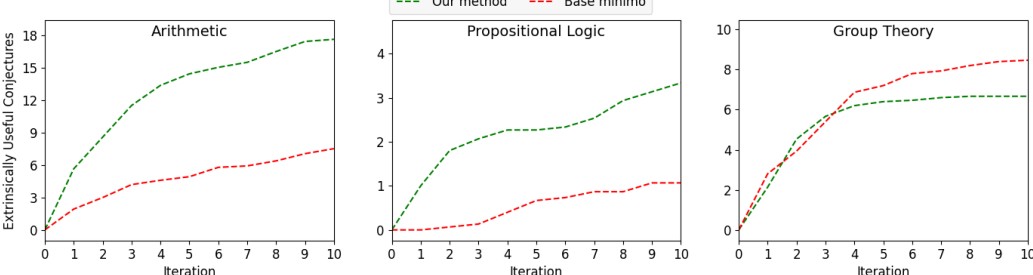

Figure 3: Extrinsic Evaluation: Number of deduplicated useful theorems per iteration, as determined by GPT-4.1 as a judge and proved by an SMT solver.

**(RQ4) Usefulness training is necessary.**  This experiment evaluates how our usefulness training signal affects performance (Figure 4). We focus here on the domain of arithmetic, though the same pattern holds in the other domains. As shown in Figure 4, if training is omitted, the system performs markedly worse: extrinsically, fewer theorems are judged to be useful by LLM-as-a-judge and SMT solver. This demonstrates the importance of training for updating the conjecturer with usefulness feedback steers it toward generating conjectures that are genuinely valuable for future proofs.

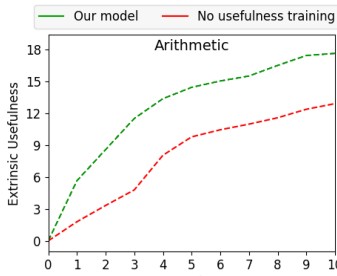

Figure 4: Comparison showing the necessity of usefulness training.

## 5 CONCLUSION AND DISCUSSION

We studied automated conjecturing from minimal axioms as a prerequisite to scalable, self-improving theorem proving. While prior self-play systems such as (Poesia et al., 2024) use difficulty (low proof log-probability) as the sole training signal, we argued that difficulty alone is insufficient for theory building. We introduced a usefulness-aware self-play framework that evaluates conjectures by their downstream impact: whether they are actually reused in subsequent proofs and whether their inclusion increases the success likelihood of proving other targets. This dual criterion operationalizes the intuition that valuable theorems function as *compression primitives* for mathematics, turning isolated wins into reusable structure. Integrated into the self-play loop, the metric selects, promotes, and trains on lemmas that reshape future proof search through usefulness.

Across arithmetic, propositional logic, and group theory, UseFor steadily increases both the intrinsic reuse of conjectured theorems and the extrinsic usefulness of its discoveries. Performance improves over successive iterations, with the system progressing from "easy" lemmas to conjectures requiring deeper proofs. Ablation studies show that training both the prover and conjecturer with usefulness feedback is necessary: removing either sharply reduces both intrinsic and extrinsic metrics. To-

gether, these findings confirm that usefulness-aware self-play can build coherent and cumulative theories directly from axioms.

**Limitations and future work.** In order to avoid the risk of data contamination, our study focuses on relatively small models, limited axioms, and fixed search budgets. Scaling to richer foundations (e.g., Lean, Isabelle) and larger models remains an open but promising direction. Our method provides potential for synthetic data generation. We have previously seen that training on synthetic data to finetune LLM-based provers such as in Dong & Ma (2025), which built on Minimo's methodology, has led to strong results. Incorporating a notion of "usefulness", as explored in our work, could further enhance the data quality and diversity, thus strengthening the provers. In addition, our method offers the potential for lemma generation at larger scale, allowing for the model to have access to powerful and useful lemmas for use for its own theorem proving. However, applying approaches like UseFor on bigger pretrained models for conjecturing, such as LLMs, brings the novel risk of data contamination. In this case, benchmarking a conjecturing and discovery model becomes extremely difficult, as it is likely that the desired theorems are contained within the training set, even if the model is constructing the theorem indirectly. Therefore, future work should consider how to mitigate such data leakage concerns in the setting where LLMs are used for conjecturing new mathematical theorems.

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

## A AXIOMS

We now provide all the axioms for the three domains considered in our experiments in Section 4. They are taken from the Minimo paper (Poesia et al., 2024) and formalized in the Peano languages (Poesia & Goodman, 2023).

**Arithmetic**

```
= : [nat -> nat -> prop].
nat : type.

z : nat.
s : [nat -> nat].
o : nat.

+ : [nat -> nat -> nat].
* : [nat -> nat -> nat].

o_s : (= o (s z)).
```

```
594
595    +_z : [('n : nat) -> (= (+ 'n z) 'n)].
596    +_s : [('n : nat) -> ('m : nat) -> (= (+ 'n (s 'm)) (s (+ 'n 'm)))].
597
598    *_z : [('n : nat) -> (= (* 'n z) z)].
599    *_s : [('n : nat) -> ('m : nat) -> (= (* 'n (s 'm)) (+ 'n (* 'n 'm)))].
600
601    nat_ind : [('p : [nat -> prop]) -> ('p z) -> [('n : nat) ->
602               ('p 'n) -> ('p (s 'n))] -> [('n : nat) -> ('p 'n)]].
603
604    #backward nat_ind.
605    #forward +_z ((+ 'n z) : nat).
606    #forward +_s ((+ 'n (s 'm)) : nat).
607    #forward *_z ((* 'n z) : nat).
608    #forward *_s ((* 'n (s 'm)) : nat).
```

**Propositional logic**

```
prop : type.

false : prop.

/* Connectives */
not : [prop -> prop].
and : [prop -> prop -> prop].
or : [prop -> prop -> prop].
iff : [prop -> prop -> prop].

/* Introduction rule for conjunction */
#backward and_i.
and_i : [('P : prop) -> ('Q : prop) -> 'P -> 'Q -> (and 'P 'Q)].
/* Elimination rules for conjunction */
#forward and_el ('_ : (and 'P 'Q)).
and_el : [('P : prop) -> ('Q : prop) -> (and 'P 'Q) -> 'P].
#forward and_er ('_ : (and 'P 'Q)).
and_er : [('P : prop) -> ('Q : prop) -> (and 'P 'Q) -> 'Q].

/* Introduction rules for disjunction */
#backward or_il.
or_il : [('P : prop) -> ('Q : prop) -> 'P -> (or 'P 'Q)].
#backward or_ir.
or_ir : [('P : prop) -> ('Q : prop) -> 'Q -> (or 'P 'Q)].
/* Elimination rule for disjunction */
#backward or_e infer infer infer infer subgoal subgoal.
or_e : [('P : prop) -> ('Q : prop) -> ('R : prop) ->
        (or 'P 'Q) -> ['P -> 'R] -> ['Q -> 'R] -> 'R].

/* Introduction rule for negation */
#backward not_i.
not_i : [('P : prop) -> ['P -> false] -> (not 'P)].
/* Elimination rule for negation */
not_e : [('P : prop) -> (not 'P) -> 'P -> false].
#backward exfalso.
exfalso : [false -> ('P : prop) -> 'P].

/* Introduction rules for equivalence */
#backward iff_i.
iff_i : [('P : prop) -> ('Q : prop) -> ['P -> 'Q] -> ['Q -> 'P] -> (iff '
    P 'Q)].
/* Elimination rules for equivalence */
#forward iff_el ('_ : (iff 'P 'Q)).
iff_el : [('P : prop) -> ('Q : prop) -> (iff 'P 'Q) -> ['P -> 'Q]].
#forward iff_er ('_ : (iff 'P 'Q)).
iff_er : [('P : prop) -> ('Q : prop) -> (iff 'P 'Q) -> ['Q -> 'P]].
```

```
/* Excluded middle */
#forward em.
em : [('P : prop) -> (or 'P (not 'P))].
```

**Group theory**

```
= : [('t : type) -> 't -> 't -> prop].

G : type.

op : [G -> G -> G].
id : G.

/* Associativity */
#forward op_assoc ((op (op 'a 'b) 'c) : G).
op_assoc : [('a : G) -> ('b : G) -> ('c : G) ->
          (= (op (op 'a 'b) 'c) (op 'a (op 'b 'c)))].

/* Commutativity */
#forward op_comm ((op 'a 'b) : G).
op_comm : [('a : G) -> ('b : G) -> (= (op 'a 'b) (op 'b 'a))].

/* Identity */
#forward id_l.
id_l : [('a : G) -> (= (op id 'a) 'a)].

/* Inverse */
inv : [G -> G].
#forward inv_l.
inv_l : [('a : G) -> (= (op (inv 'a) 'a) id)].
```

## B    EXTRINSIC EVALUATION

In order to perform extrinsic evaluation, we run 5 iterations of our extrinisc evaluation pipeline, and take the average of the 5 results in order to mitigate variance from different runs of LLM evals. Our extrinsic evaluation pipeline consists of two steps: usefulness checking (Appendix B.1), deduplication (Appendix B.2), and SMT solving. In usefulness checking, we prompt the model concurrently on all conjectures generated by the model and keep the ones marked as useful by the LLM. As we are concurrently requesting for usefulness, we are likely to get a large amount of duplicate conjectures. We therefore make a second pass, calling the model on the useful conjectures to deduplicate them, keeping only sufficiently different theorems so as to get more reasonable results. Finally, we leverage the Z3 SMT solver (De Moura & Bjørner, 2008) to automatically prove the remaining conjectures and count only the proven ones. We found Z3 to be highly effective in proving these conjectures, as they are derived from axioms.

In the specific case of group theory, we noticed the variance in LLM evaluations was significantly higher than other domains, and the LLM had a very high rate of returning false problems. We solved this by running the SMT solver first, and giving a custom deduplication prompt (Appendix B.4) with examples for group theory.

### B.1    USEFULNESS CHECKING PROMPT

```
You are tasked to judge whether a given lean theorem could be considered
    useful for an automatic theorem prover to have among its known
    theorems.
This theorem prover has only access to the following axioms and known
    theorems:
```
```
{known_theorems}
As well as access to the `rfl` and `rewrite` commands
```

```
Here is the theorem you are to evaluate
```lean4
{generated_conjecture}
```
Think through the problem step by step. Translate the problem into
    natural language, then think of what the possible uses of the theorem
     could be, whether it's obviously true and whether it means something
     .
On the last line, say either USEFUL or NOT USEFUL and nothing else.
```

## B.2   DEDUPLICATION PROMPT

```
I have a set of lean theorems, some of which are very similar to each
    other. I want to use them as tactics for proof generation.
Please remove the duplicates, so that I can have a list of only unique
    theorems.
For example, the following four theorems would be duplicates of each
    other:
```lean4
theorem problem1 : (v0 : Nat) -> v0 * 1 = v0
theorem problem2 : (v0 : Nat) -> (v1 : Nat) -> v1 * 1 = v1
theorem problem3 : (v0 : Nat) -> (v1 : Nat) -> (v2 : v0 = v1) -> v1 * 1 =
     v1
theorem problem4 : (v0 : Nat) -> v0 * (Nat.succ 0) = v0
```
The inclusion of an extra variable in problem 2 doesn't change the fact
    that the result is exactly the same, and the different names for the
    variable doesn't affect the result.
Problem 3 introduces an irrelevant hypothesis, which doesn't get used in
    the theorem, and the conclusion is still the same.
The last one is a trivial result of the others, as 1 is defined as Nat.
    succ 0 in this case.
Here is my list of theorems for you to remove duplicates for.
{}
I also have attached an explanation for why each could be useful for a
    theorem prover.
{}
Think it through step by step, and then return the list of unique
    theorems from this list in a list format inside of a ```lean4``` code
     block. Make sure your answer is inside the very last lean codeblock.
     Please make sure to repeat the theorems exactly as I wrote them.
```

## B.3   DISTINGUISHING UNIQUE CONJECTURES BETWEEN MODELS

As seen in Appendix B, each experiment involves 5 iterations in order to get sets of extrinsically useful conjectures, each of which gives a slightly different set of extrinsically useful conjectures. In order to determine which conjectures were only conjectured by one model, we first union the sets we obtained from each of the 5 iterations we obtained for each model. From this, we get a resulting set of all conjectures considered at some point useful by the LLM for each model we wish to compare. We then perform deduplication (Appendix B.2), as there might be equivalent conjectures between iterations. We then union our two resulting sets together and call the LLM with our distinguishing prompt, and determine which model conjectured each of the resulting conjectures, with the following prompt:

```
I have the following list of lean theorems. I would like you to select
    all 'unique' lean4 theorems, that is ones that have no other theorem
    that is semantically equivalent in the list.
For example, the following four theorems would be duplicates of each
    other:
```lean4
theorem problem1 : ((v0 : Group) -> (v1 : (v0 = (v0 * (1ˆ{-1}))))) ->
    ((1ˆ{-1}) = 1))
theorem problem2 : ((v0 : Group) -> (v1 : Group) -> ((1ˆ{-1}) = 1))
```

```
theorem problem3 : ((v0 : Group) -> ((1^{-1}) = 1))
theorem problem4 : ((v0 : Group) -> (1 = (1^{-1})))
```
Problem 1 introduces an irrelevant hypothesis as compared to problem 3,
    as it makes no mention of v0 in its final claim. Therefore, these two
     problems are duplicates of each other.
Problem 2 is a similar case to problem 1: It introduces an extra variable
    , but does nothing with it. This is irrelevant, and makes for the
    same problem.
Problem 4 is the same as problem 3, but is flipped. As we are running
    this using rw, we can simply call this problem in the inverse
    direction, so these two lemmas are the same.

Think this step by step, and then give your answer in a ```lean4 ``` code
    block. Make sure to write the theorem exactly as written.
Here are the lean4 theorems:

### B.4 GROUP THEORY SPECIFIC PROMPTS

I have a set of lean theorems, some of which are very similar to each
    other. I want to use them as lemmas for proof generation.
Please remove the duplicates, so that I can have a list of only unique
    theorems.
For example, the following four theorems would be duplicates of each
    other:
```lean4
theorem problem1 : ((v0 : Group) -> (v1 : (v0 = (v0 * (1^{-1}))))) ->
    ((1^{-1}) = 1))
theorem problem2 : ((v0 : Group) -> (v1 : Group) -> ((1^{-1}) = 1))
theorem problem3 : ((v0 : Group) -> ((1^{-1}) = 1))
theorem problem4 : ((v0 : Group) -> (1 = (1^{-1})))
```
Problem 1 introduces an irrelevant hypothesis as compared to problem 3,
    as it makes no mention of v0 in its final claim. Therefore, these two
     problems are duplicates of each other.
Problem 2 is a similar case to problem 1: It introduces an extra variable
    , but does nothing with it. This is irrelevant, and makes for the
    same problem.
Problem 4 is the same as problem 3, but is flipped. As we are running
    this using rw, we can simply call this problem in the inverse
    direction, so these two lemmas are the same.

In this case, our final result would likely be:
```lean4
theorem problem3 : ((v0 : Group) -> ((1^{-1}) = 1))
```

Here is my list of theorems for you to remove duplicates for.
{}
I also have attached an explanation for why each could be useful for a
    theorem prover.
{}
Think it through step by step, and then return the list of unique
    theorems from this list in a list format inside of a ```lean4``` code
     block. Make sure your answer is inside the very last lean codeblock.
     Please make sure to repeat the theorems exactly as I wrote them.

### B.5 EXAMPLES OF EXTRINSICALLY USEFUL CONJECTURES

Table 1 highlights representative conjectures that our evaluation judged to be extrinsically useful across three domains. These serve as concrete examples of the kinds of results UseFor is capable of producing. As an illustration, UseFor produces a 5-step proof of the first propositional-logic

conjecture in Table 1, using only the base axioms. However, given the tactic `iff_elim`, which reduces an equivalence to two implications, together with the axioms

$$False \implies P, \tag{1}$$
$$P \wedge Q \implies P, \tag{2}$$

UseFor found the following 5-step proof:

    1.Split the problem into cases:          by `iff_elim`

    − Case 1: $False \implies P \wedge False$

      2.introduce $False$ into hypothesis context

      3.$False \implies P \wedge False$          by (1)

    − Case 2: $P \wedge False \implies False$

      4.introduce $P \wedge False$ into hypothesis context

      5.$P \wedge False \implies False$          by (2)

This example demonstrates how UseFor produces lemmas that apply broadly and compress multiple reasoning steps into a single inference step in practice. This ability provides a crucial advantage in Monte Carlo Tree Search, where the search space expands exponentially with depth.

We remark that these proven conjectures are also observed to be very important to the prover in future iterations. For instance, $P \implies \neg\neg P$ and $1^{-1} = 1$ often serve as powerful shortcuts, condensing multi-step reasoning into a single step and thereby streamlining longer proofs.

| **Arithmetic** | **Propositional Logic** | **Group Theory** |
|---|---|---|
| $\forall x \in \mathbb{N}, x(x^2 + 1) = x + x^3$ | $False \iff (P \wedge False)$ | $1^{-1} = 1$ |
| $2x = 0 \implies x = 0$ | $P \implies \neg\neg P$ | |
| $\forall x \in \mathbb{N}, x * 1 = x$ | $P \iff P$ | $\forall x \in G, x \cdot x = x \implies x = 1$ |

Table 1: Representative conjectures judged extrinsically useful across three considered domains.

## C  TRAINING DETAILS

We instantiate our GPT-2 model with 8.45M parameters, with 8 layers, 8 attention heads, a hidden size of 512, 2048 feed forward, a vocabulary size of 128, absolute positional embeddings, and with a maximum context of 1024. We train the language model after every iteration with 2000 steps of the AdamW optimizer (learning rate of $1e - 4$). Monte Carlo Tree Search is done with a max expansion of 1000. We train each model over 10 iterations with 200 conjectures per iteration, and run on between 1 and 2 H100 80GB GPUs. An average run takes between 12-24 hours on one GPU.

## D  THE USE OF LARGE LANGUAGE MODELS

Large Language Models (LLMs) were used to support writing, revision, and other text-focused tasks, such as improving clarity, refining grammar and style, and assisting with the organization of written content.

