# OpenReview forum: "Usefulness-driven Learning of Formal Mathematics"
_ICLR.cc/2026/Conference — Submitted to ICLR 2026_

### Official Review · Reviewer_kAWp · 2025-10-26

**Soundness:** 2
**Presentation:** 3
**Contribution:** 3
**Rating:** 6
**Confidence:** 3

**Summary:**

The paper introduces UseFor, a usefulness-driven framework for automated discovery of formal theorems. Rather than ranking conjectures only by “difficulty” (low proof likelihood) as in Minimo, UseFor promotes conjectures that (i) are actually used in downstream proofs and (ii) increase the prover’s success likelihood on those targets. The method integrates this metric into a self-play loop (conjecturing & proving), adds triviality filtering and novelty bias, and evaluates across arithmetic, propositional logic, and group theory. Empirically, UseFor yields more reusable lemmas and higher “usefulness” (LLM-as-judge + SMT-proved) than difficulty-only baselines; e.g., in arithmetic they report 2.9× extrinsic usefulness over Minimo.

**Strengths:**

- **Clear, principled objective**: The dual criterion (usage + likelihood improvement) operationalizes “theorem usefulness” in a tractable way that aligns with theory building, not just local hardness. The formulation and scoring rule are precise.
- **Thoughtful integration into self-play**: UseFor plugs into Minimo’s loop but changes the selection signal, and adds triviality filtering and novelty bias to avoid vacuous lemmas and near-duplicates—practical details that matter for stability.
- **Compelling empirical story**: Multiple domains, explicit intrinsic vs. extrinsic metrics, and a meaningful ablation. The results have demonstrated very meaningful improvements in mathematical discovery.

**Weaknesses:**

- **Limited Technical Details**: Obviously, this is a good follow-up work for [1]. However, the authors provided little recap of previous technical details, making the paper somewhat difficult to read. The authors should have built the setup more clearly. For example, what kind of LM settings are you using to prevent the contamination issue?
- **Limited Coverage in Mathematics**: The work has good motivations for self-evolving in mathematical discovery, but the axioms and topics are limited as they are also very similar, i.e., heavily relying on the type theory structure.
- **Reporting Clarity**: The paper asserts super-linear growth in usefulness evaluations; please include exact values with error bars per iteration and statistical tests.
- **Scale and generality**: Results are on small models and Peano-style domains; applicability to Lean/Isabelle with larger models remains open, and the paper itself flags contamination risks when scaling with pre-trained LLMs. A small-scale Lean pilot (even with tight libraries) or cross-formalism transfer test would strengthen generality claims.

[1] Poesia, Gabriel, et al. "Learning formal mathematics from intrinsic motivation." Advances in Neural Information Processing Systems 37 (2024): 43032-43057.

**Questions:**

- Typo: Line 334: domain --> domains

---

> ### Author Response · Authors · 2025-11-21
>
> We sincerely thank the reviewer for their constructive feedback and will address each of their questions below. We are also updating the paper to incorporate all necessary revisions and will upload the revised version to OpenReview in a few days.
>
> **W1: Can you provide more technical details on Minimo? For example, what LM settings did you use?**
>
> Yes, we will include more details in the experimental section. Currently, section 3.1 has an overview of Minimo’s high-level approach. The experimental setup is the same as Minimo[1], with the exception that we ran each experiment over 10 iterations, whereas the original paper ran for 5 iterations. Otherwise, the only difference is the core technical approach. Our LM settings are also the same as the original paper, with MCTS proof search with 1000 expansions proving 200 conjectures per iteration.
>
> **W2: Why is it useful to focus on only a limited set of mathematical domains closely related to type theory structure?**
>
> We chose to match the mathematical domains used with those of the Minimo paper [1] in order to properly compare our results with Minimo. We hope to show through our multiple domains that our method could be extended to other domains if one wished. Dependent type theory is widely used in order to write formal proofs and theorems, and has been the backbone of much work in applying AI to mathematics, such as with AlphaProof [3].
>
> **W3: Can you clarify the growth of your usefulness metric?**
>
> Claiming “super-linear growth” was a typo. We will adjust the statement to “The growth of this metric empirically suggests that UseFor can generate theorems that are regarded as useful in the real-world by LLM-as-judge”.
>
> **W4: What is the motivation for having a small model?**
>
> We focused on smaller models trained from scratch for an important scientific reason: to avoid data contamination and ensure that all generated conjectures are genuine discoveries. We adopted this experimental setup directly from Minimo, which also enables direct comparison.
>
> We acknowledge that extending our approach to larger models is a valuable future direction. For example, STP [2] successfully scaled the principles in Minimo to LLM-based provers. Given our improvements over Minimo, we expect our approach to similarly lead to positive results.
>
> [1] Poesia, Gabriel, et al. "Learning formal mathematics from intrinsic motivation." NeurIPS 2024 \
> [2] Dong, Kefan, and Tengyu Ma. "Stp: Self-play llm theorem provers with iterative conjecturing and proving." ICML 2025. \
> [3] Hubert, Thomas, et al. "Olympiad-level formal mathematical reasoning with reinforcement learning." Nature (2025): 1-3.

---

> > ### Author Response · Authors · 2025-11-27
> >
> > We thank you for your patience and have now uploaded a revised version of the paper. We have colored our revisions in blue. Below is a list of summarizing our revisions, with a one-to-one correspondence with our rebuttal above:
> >
> > - **W1**: We have added a statement in Section 4.1 describing model parameters, as well as an extended description of model details in the appendix (Lines 389-391).
> >
> > - **W2**: We have revised the paper to mention that we are using the same domains as Minimo (Lines 383-385)
> >
> > - **W3**:  We have adjusted the statement accordingly in RQ3 (Lines 438-439).
> >
> > - **W4**:  We have added discussion of why we trained a small model from scratch to avoid data contamination, and how our work could be extended to larger models in our future works section (Lines 491-500).
> >
> > To ensure we have ample time to address any outstanding concerns you may have, we would kindly inquire if our response has addressed your concerns. We remain fully available to answer any further questions or provide additional clarification. We would be very grateful if you would consider these clarifications in your final evaluation.

---

### Official Review · Reviewer_SsWm · 2025-11-01

**Soundness:** 3
**Presentation:** 1
**Contribution:** 3
**Rating:** 4
**Confidence:** 4

**Summary:**

This paper introduces UseFor, which builds upon prior formal mathematics work Minimo. While Minimo engages in a self-play loop by rewarding hard-but-not-impossible conjecturing, UseFor seeks to measure usefulness of lemmas for a benchmark set $\mathcal B$ of theorems. UseFor also adds library learning and triviality checks. They demonstrate that UseFor is able to increasingly use past theorems and scores higher than Minimo on an LLM-as-judge extrinsic usefulness score. They establish that UseFor beats a baseline that is Minimo but with library learning.

**Strengths:**

Originality: I think this is a fantastic idea to try and is novel. Intuitively, it makes a great deal of sense that theorem utility in proving other things is critical for deciding what is interesting to include as a theorem. The integration of library learning and triviality checks also make a great deal of sense, and it’s useful to integrate these ideas.

Quality: The treatment of related work and motivation for these ideas is very thorough. The set of RQs for the experiments section is well-motivated and for the most part executed well.

Clarity: For the most part, I found this very easy to follow. Everything was well-motivated.

Significance: the methodological contribution here is I think of considerable interest in this growing field of open-ended learning & formal mathematics. The experiments present good evidence.

**Weaknesses:**

While this is for the most part very clearly written, I cannot seem to find certain really critical details that would be important for both reproducibility and for interpreting the results. In particular, how is the benchmark set of targets $\mathcal B$ set? If this is something that is created and dynamically updated by the algorithm — if so, how? How is it seeded? How is it updated to reflect the changing capabilities of the model (if indeed it’s supposed to be “hard”)? These seem like important details, not at all trivial. If $\mathcal B$ is something that is not created by the algorithm, but rather is provided, that puts UseFor on a very different footing relative to Minimo, calling all the comparisons into question.

This work also adopts LLM-as-judge for an extrinsic usefulness measure. Minimo has another measure for how useful the play process is, namely, whether it gets better at proving a held out set of theorems provided by people. It would be really helpful to understand what this is and how it relates to the LLM-as-judge metric.

In answering RQ3, the paper claims super-linear growth. I’m not seeing that from the metric, though perhaps I’m misunderstanding something.

It would also be helpful to see the No usefulness training baseline for answering RQ2, as that Minimo didn’t use library learning seems a bit orthogonal to the point you’re making for that plot.

**Questions:**

How was $\mathcal B$ set up and maintained? I’d be keen to increase my score if this, in particular, is addressed well.

More minor but I think still critical for understanding the method: how is $\mathcal H_i$ (I’m assuming this is a “hard” subset of conjectures) maintained throughout training? Is there a set of “inner loop” successive subsamplings L_i made, in between model updates? If so, how many? If not, how is it that all conjectures eventually receive usefulness credit?

Can we have some error analysis for what’s going on with Group Theory relative to the other domains? The reversal in performance relative to Minimo is striking there.

Does No usefulness training have the triviality check? It’s not 100% clear to me whether a simple hardness criterion doesn’t screen those out.

---

> ### Author Response · Authors · 2025-11-21
>
> We sincerely thank the reviewer for their constructive feedback and will address each of their questions below. We are also updating the paper to incorporate all necessary revisions and will upload the revised version to OpenReview in a few days.
>
> **W1 & Q1: What is the benchmark set $\mathcal{B}$ in your self-play loop?**
>
> In our self-play loop (Section 3.2.2), the benchmark set $\mathcal{B}$ is defined as the subset of hard conjectures that are generated and successfully proven during the current iteration. Thus, $\mathcal{B}$ is created internally in our self-play loop and updated dynamically at each iteration to reflect the model’s evolving capabilities. This design mirrors that of Minimo[1], in the sense that neither method depends on any external benchmark other than the axioms. We will revise the paper to make this clearer.
>
> **W2: Minimo benchmarked its models on a set of theorems to prove. How does this relate to the LLM-as-judge metric?**
>
> Both evaluations serve the same high-level purpose: to ground the self-play process and demonstrate that improvements in internal metrics translate into gains on an external measure. In Minimo, the objective is theorem proving, so performance on a held-out proof benchmark provides an extrinsic assessment of proving ability. In our work, the objective is to generate useful conjectures, so we use the LLM-as-judge metric as an external measure of conjecture quality. In both cases, the aim is to ensure that models are not merely improving within the self-play environment but also exhibit genuine, externally validated progress. We will clarify this connection in the paper.
>
> **W3: Can you clarify the growth of your usefulness metric?**
>
> Claiming “super-linear growth” was a typo. We will adjust the statement to “The growth of this metric empirically suggests that UseFor can generate theorems that are regarded as useful in the real-world by LLM-as-judge”.
>
> **W4: It would be helpful to see the No-usefulness-training baseline for addressing RQ2, since Minimo not using library learning seems orthogonal to the point you’re making in that plot.**
>
> We will add this baseline to Figure 2. The results we obtained are shown below, demonstrating that our method consistently achieves higher log probability than the baseline. We also note that the two curves are relatively close. This is expected: during proof search, the model typically uses a lemma for one or a few steps of the proof rather than shifting the entire proof trajectory.
>
> |          Iteration              | 0      | 1      | 2     | 3     | 4     | 5     | 6     | 7     | 8     | 9     |
> |------------------------|--------|--------|-------|-------|-------|-------|-------|-------|-------|-------|
> | Logprob (Our method)   | -17.42 | -10.05 | -7.11 | -5.94 | -4.76 | -5.28 | -4.88 | -4.26 | -5.32 | -4.70 |
> | Logprob (No usefulness training)     | -17.53 | -10.92 | -7.37 | -7.06 | -6.89 | -5.82 | -5.81 | -6.75 | -5.67 | -6.01 |
>
> **Q2: What is $\mathcal{H}_i$? How is it maintained throughout training? Is $L_i$ resampled during an iteration?**
>
> $\mathcal{H}_i$ denotes the cumulative set of all theorems proven up to iteration $i$. During usefulness evaluation, we subsample a subset of $\mathcal{H}_i$ to form the theory $L_i$ used for testing. Within a single iteration, $L_i$ is not resampled. However, any theorem in $\mathcal{H}_i$ that is not included in $L_i$ at iteration $i$ remains part of the global cumulative set and can be incorporated into future iterations. This ensures that all previously proven theorems remain available to the process, even if only a subset is used at any given iteration.
>
> **Q3: What is happening with the model when it comes to group theory?**
>
> As seen in Figure 3, our model doesn’t seem to provide much improvement over Minimo’s method on average. However, if we union all runs, and use an LLM to perform deduplication, we are able to find that we generated 23 conjectures that were not semantically equivalent to any generated by Minimo, with Minimo generating 21 that were not generated by our model, and a total of 64 shared between the two models. Therefore, we appear to generate less in quantity on average, but we still remain diverse, and the generated theorems between our model and Minimo are complementary.
>
>
> **Q4: Does the “No usefulness training” from Figure 2 include trivial filtering as described in section 3.2.3?**
>
> We enabled all the improvement techniques in Section 3.2.3 for the “No usefulness training” version, including triviality filtering. This means the only variable is whether the usefulness training is enabled, facilitating a clean comparison.
>
> [1] Poesia, Gabriel, et al. "Learning formal mathematics from intrinsic motivation." NeurIPS 2024

---

> > ### Author Response · Authors · 2025-11-27
> >
> > We thank you for your patience and have now uploaded a revised version of the paper. We have colored our revisions in blue. Below is a list of summarizing our revisions, with a one-to-one correspondence with our rebuttal above:
> >
> > - **W1, Q1**: We have revised section 3.2.1 and 3.2.2 to clarify what the benchmark set B_i is and that it is generated internally in our self-play training loop (Lines 251-252, 292-294).
> >
> > - **W2**: We have added a paragraph describing Minimo’s evaluation methods and how they compare to our methods in Section 4.1 (Lines 369-373).
> >
> > - **W3**: We have adjusted the statement accordingly in RQ3 (Lines 438-439).
> >
> > - **W4**: We have added the data to Figure 2, and added the explanation given in our rebuttal to our analysis in RQ2 (Lines 432-434).
> >
> > - **Q2**: We have revised section 3.2.2 to clarify what the benchmark set H_i is and how it is generated, and how L_i is sampled from H_i  (Lines 292-294).
> >
> > - **Q3**: We have added the analysis given in the rebuttal to RQ3 (Lines 440-446).
> >
> > - **Q4**: We have added a paragraph to Section 4.1 detailing what the no usefulness training baseline entails, including the fact that the baseline does use triviality filtering (Lines 375-380).
> >
> > To ensure we have ample time to address any outstanding concerns you may have, we would kindly inquire if our response has addressed your concerns. We remain fully available to answer any further questions or provide additional clarification. We would be very grateful if you would consider these clarifications in your final evaluation.

---

### Official Review · Reviewer_JUi9 · 2025-11-01

**Soundness:** 2
**Presentation:** 3
**Contribution:** 2
**Rating:** 6
**Confidence:** 4

**Summary:**

This paper proposes UseFor, a framework for proposing theorems that are useful and can be used when training usefulness-driven automated math. Specifically, the work formalizes the notion of usefulness as a criterion for both usage and improvement. The work is integrated into the Minimo framework, a framework for automatically learn to prove or disprove conjectures. As part of the integration, the authors use usefulness instead of difficulty as a criterion for selecting new conjectures. Moreover, various new techniques are introduced in the method, including stabilization techniques such as filtering and novelty bias. The approach is evaluated on three different mathematical domains.

**Strengths:**

- This kind of work, incorporating interestingness info into a pipeline for automatic conjecture discovery and theorem proving, is interesting and an important direction in AI math discovery

- The paper shows an interesting and pretty simple way of incorporating a notion of interestingness into an automatic math learning loop

**Weaknesses:**

- One aspect of the difference between the proposed extension to Minimo and the original Minimo work is that the proposed method is driven by a benchmark set; I.e., a benchmark is used for computing the interestingness score. In contrast, Minimo is not driven by any external benchmark theorems. This is not negative in itself, but a clear difference to the original work, which is important to highlight clearly in the paper.

- The benchmark setup is not clearly described in the paper. That is, Section 3.2.1 describes how the interesting value is computed using a benchmark set. However, it is unclear in the experimental evaluation where this benchmark set is coming from or how it is integrated into the framework.

- As already noted by the authors, the experimental setup contains rather small models

**Questions:**

- Are the three mathematical domains and their definitions identical to the original Minimo paper, or are there any differences?

- Lines 412 to 413 say "The growth of this metric is super-linear, suggesting that UseFor not only maintains but amplifies its ability to generate theorems a mathematician would regard as useful." This is quite a strong statement, since I cannot see how the proposed approach gives any guarantee that the paper's notion of interestingness corresponds to what a mathematician would regard as useful. Please clarify this statement if possible.

- For the Minimo version with "No usefulness training" (line 431), how does it select the proven conjectures? Or is it including all without filtering?

---

> ### Author Response · Authors · 2025-11-21
>
> We sincerely thank the reviewer for their constructive feedback and will address each of their questions below. We are also updating the paper to incorporate all necessary revisions and will upload the revised version to OpenReview in a few days.
>
> **W1 and W2: Your interestingness metric depends on a benchmark set. Where is this benchmark set from for your self-play loop?**
>
> It is correct that in Section 3.2.1, the general form of our interestingness metric is defined with the help of a benchmark set. However, to clarify, when we instantiate our metric in the self-play loop in Section 3.2.2, we set the benchmark set to be the “hard” subset of conjectures generated and proven in the current iteration. This benchmark set is generated entirely internally in the self-play loop and does not use any external information. Therefore, in our experimental evaluation, we do not need to specify an external benchmark set. This also makes our setup the same as Minimo[1], in that both do not rely on any external benchmark theorems. We will revise our paper to clarify this.
>
> **W3: A limitation of your evaluation is that you only focus on smaller models.**
>
> We focused on smaller models trained from scratch for an important scientific reason: to avoid data contamination and ensure that all generated conjectures are genuine discoveries. We adopted this experimental setup directly from Minimo, which also enables direct comparison.
>
> We acknowledge that extending our approach to larger models is a valuable future direction. For example, STP [2] successfully scaled the principles in Minimo to LLM-based provers. Given our improvements over Minimo, we expect our approach to similarly lead to positive results.
>
>
> **Q1: Are the three mathematical domains and their definitions identical to the original Minimo paper, or are there any differences?**
>
> Our paper uses identical axioms to the original Minimo paper. The full list will be included in the appendix section, which will make it clear that this is the case.
>
> **Q2: Can you adjust your statement on Lines 412 to 413?**
>
> We acknowledge that this statement might sound inappropriate. We will revise this statement to be “The growth of this metric empirically suggests that UseFor can generate theorems that are regarded as useful in the real-world by LLM-as-judge”.
>
> **Q3: For the Minimo version with "No usefulness training" (line 431), how does it select the proven conjectures? Or is it including all without filtering?**
>
> We enabled all the improvement techniques in Section 3.2.3 for the “No usefulness training” version, including triviality filtering. This means the only variable is whether the usefulness training is enabled, facilitating a clean comparison.
>
> [1] Poesia, Gabriel, et al. "Learning formal mathematics from intrinsic motivation." NeurIPS 2024 \
> [2] Dong, Kefan, and Tengyu Ma. "Stp: Self-play llm theorem provers with iterative conjecturing and proving." ICML 2025.

---

> > ### Author Response · Authors · 2025-11-27
> >
> > We thank you for your patience and have now uploaded a revised version of the paper. We have colored our revisions in blue. Below is a list of summarizing our revisions, with a one-to-one correspondence with our rebuttal above:
> >
> > - **W1 and W2**: We have revised section 3.2.1 and 3.2.2 to clarify what the benchmark set B_i is and that it is generated internally in our self-play training loop (Lines 251-252, 292-294). We have also reiterated in our experimental setup that no external benchmark was used (Lines 385-387).
> >
> > - **W3**: We have added discussion of how our work can be extended to large models in our future works section (Lines 491-500).
> >
> > - **Q1**: We have added a sentence clarifying we use the same axioms as Minimo (Lines 385)
> >
> > - **Q2**: We have adjusted the statement accordingly in RQ3 (Lines 438-439).
> >
> > - **Q3**: We have added a paragraph to Section 4.1 detailing what the no usefulness training baseline entails, including the fact that the baseline does use triviality filtering (Lines 375-380).
> >
> > To ensure we have ample time to address any outstanding concerns you may have, we would kindly inquire if our response has addressed your concerns. We remain fully available to answer any further questions or provide additional clarification. We would be very grateful if you would consider these clarifications in your final evaluation.

---

### Official Review · Reviewer_kBd3 · 2025-11-02

**Soundness:** 2
**Presentation:** 2
**Contribution:** 2
**Rating:** 4
**Confidence:** 3

**Summary:**

The paper introduces UseForm, a novel framework to train AI theorem provers to generate theorems that are useful and can serve as building blocks for proving more advanced subsequent theorems. Building on top of Minimo which starting from only axioms, iteratively train conjecturers to propose useful formal statements and provers that explicitly reuse them when generating formal proofs, the authors replaces difficulty-based training with a usefulness-aware self-play loop. The usefulness metric measures how much adding a lemma improves success likelihood and reuse in future proofs. The authors evaluate the proposed approach across three mathematical domains: arithmetic, propositional logic, and group theory, with two metrics: intrinsic usefulness, which measures how often the trained provers reuse theorems, and extrinsic usefulness, judged by a state-of-the-art LLM and external provers like SMT solvers. Experiment results show that UseFor significantly boosts theorem reuse and generates 2.9× more extrinsically useful theorems as judged by GPT-4.1 and Z3 solvers than Minimo.

**Strengths:**

1. The paper is well-motivated, well-written and easy to follow
2. The concept of usefulness in theorem proving is novel, shifts from proving difficult theorems to discovering useful ones.
3. The proposed usefulness-aware self-play loop is interesting.
4. The authors show consistent gains in both intrinsic (reuse) and extrinsic (human-judged) usefulness across multiple domains.

**Weaknesses:**

1. The proposed UseForm framework heavily relies on the existing Minimo framework, replacing its difficulty-based objective with a usefulness-based criterion. The improvement seems incremental.
3. While the authors demonstrate gains in the usefulness of generated proofs, it is unclear whether the UseForm framework improves or hurts proof accuracy.
2. It is unclear whether the UseForm framework can be adapted to other strong theorem-proving systems such as Seed-Prover.
4. The paper lacks comparisons with frontier theorem-proving frameworks such as Seed-Prover [1], Goedel Prover [2], and LLM-based provers such as  GPT-5, Qwen-235B, Claude Sonnet, and Grok.



[1] Chen, Luoxin et al. “Seed-Prover: Deep and Broad Reasoning for Automated Theorem Proving.” ArXiv abs/2507.23726 (2025): n. pag.

[2] Lin, Yong et al. “Goedel-Prover-V2: Scaling Formal Theorem Proving with Scaffolded Data Synthesis and Self-Correction.” ArXiv abs/2508.03613 (2025): n. pag.

**Questions:**

See weakness

---

> ### Author Response · Authors · 2025-11-21
>
> We sincerely thank the reviewer for their constructive feedback and will address each of their questions below. We are also updating the paper to incorporate all necessary revisions and will upload the revised version to OpenReview in a few days.
>
> **W1: How are these results not incremental over Minomo?**
>
> Our work represents a fundamental shift in objective rather than a minor refinement. Minimo optimizes for difficulty, which often yields "dead-end" theorems that are true but unusable. We optimize for usefulness, ensuring discoveries serve as building blocks for future proofs, a capability of cumulative learning that Minio lacks. Quantitatively, our approach results in a 2.9x increase in useful theorems compared to Minimo[1].
>
> **W2: Is the “proof accuracy” metric applicable to your work?**
>
> The metric of "proof accuracy" (pass rate on a fixed test set) does not apply to our method, as our goal is theorem discovery, not theorem proving. We use the prover as a means to filter the conjectures we generate, rather than a goal in of itself. If the prover fails, we simply discard that conjecture rather than counting it as an error. Therefore, proof accuracy is not applicable to our work; instead, we measure success by the yield, which is the total number of useful theorems we find.
>
> **W3: How can this method be adapted to existing theorem provers like Seed-Prover?**
>
> Our method can be adapted to support provers like Seed-Prover[2] or STP [3] in two practical ways. First, it can act as a lemma generator. By running our method beforehand, we can discover and list useful intermediate steps (lemmas) that give these provers the building blocks for more complex proofs. Second, it can act as a data generator. Since our system automatically produces useful theorems, this output can be used as training data to help existing provers learn better reasoning strategies for new problems.
>
> We see strong potential here because STP has already successfully scaled similar principles from Minimo to LLM-based provers. Since our method demonstrates clear improvements over Minimo, we expect these findings to translate into significant gains for these systems as well. However, fully integrating our approach into these provers requires significant effort. For instance, the evolution from Minimo to STP required an entire research paper to execute successfully. Therefore, we consider this integration an important future work item.
>
>
> **W4: How does this compare to other LLM-based theorem provers like Goedel Prover and Seed-Prover?**
>
> While our approach can be adapted to improve these theorem provers (as discussed in the previous answer), they are not directly comparable due to two key differences. First, the goal is different: our approach aims to discover useful theorems, not proving given theorems. The second difference lies in training data and model power. We train our models from scratch to avoid data contamination and ensure that all generated theorems are genuine “discoveries”. LLM-based provers, in contrast, rely on pre-trained models that may have already seen many theorems and proofs online.
>
> [1] Poesia, Gabriel, et al. "Learning formal mathematics from intrinsic motivation." NeurIPS 2024 \
> [2] Chen, Luoxin, et al. "Seed-prover: Deep and broad reasoning for automated theorem proving." arXiv preprint arXiv:2507.23726 (2025). \
> [3] Dong, Kefan, and Tengyu Ma. "Stp: Self-play llm theorem provers with iterative conjecturing and proving." ICML 2025.

---

> > ### Author Response · Authors · 2025-11-27
> >
> > We thank you for your patience and have now uploaded a revised version of the paper. We have colored our revisions in blue. Below is a list of summarizing our revisions, with a one-to-one correspondence with our rebuttal above:
> >
> > - **W3**: We have added discussion of how our work can be extended to large models like Seed-Prover in our future works section (Lines 491-500).
> >
> > - **W4**: We have emphasized that our model is trained from scratch to avoid data contamination in conjecturing (Lines 388-390, 499-500). As explained, this makes our approach incomparable to LLM-based provers, which relies on large pretrained corpora and suffers from data contamination.
> >
> > To ensure we have ample time to address any outstanding concerns you may have, we would kindly inquire if our response has addressed your concerns. We remain fully available to answer any further questions or provide additional clarification. We would be very grateful if you would consider these clarifications in your final evaluation.

---

### Author Response · Authors · 2025-12-02

Dear Chairs,

We acknowledge the decision to revert review scores and close the author-reviewer discussion due to the OpenReview security incident. We sincerely appreciate your efforts in managing ICLR during this challenging situation.

To facilitate the upcoming review process, we would like to summarize how our rebuttal addressed key concerns and strengthened our paper:
- **Benchmark set $\mathcal{B}$ (Reviewers SsWm and JUi9)**: We clarified that $\mathcal{B}$ is created internally by our algorithm and is dynamically updated to reflect the model’s evolving capabilities. This makes our work directly comparable to Minimo. **Reviewer SsWm explicitly stated they would be keen to increase their score given this clarification.**
- **Paper Objective (Reviewer kBd3)**: We resolved the reviewer’s confusion regarding the paper's scope by clarifying that our aim is to generate interesting conjectures rather than prove theorems.

We have incorporated all feedback into the latest revision of our paper on OpenReview, with itemized summaries of the changes under every review. We believe these revisions have significantly strengthened the work and hope you will consider them.

Best,
Authors

---

### Meta-Review · Area_Chair_atik · 2026-01-07

**Summary:**

Here is a summary of the  reviewers' concerns

 - Novelty and positioning still borderline (kBd3, partly JUi9). Even after rebuttal, the work can read as “Minimo + different selection signal + heuristics.” The claim that this is a “fundamental shift” is argued, but the paper may still lack a crisp delineation of what is new beyond objective swap and how much gain comes from auxiliary stabilization (filtering/novelty/library learning).
 - Benchmark-set mechanism and credit assignment clarity remains the main reproducibility risk (JUi9, SsWm). Authors say the benchmark is internally defined (hard proven conjectures from current iteration), but reviewers’ core worry persists unless the paper precisely specifies benchmark construction (seed, “hard” criterion, update rule) and the subsampling scheme used for scoring, including frequencies and sizes, and whether the scheme introduces systematic scoring bias.
 - Evaluation validity. LLM-as-judge remains weakly grounded (SsWm, kAWp, JUi9). Even with toned-down language, the extrinsic metric is still “useful per GPT-4.1 (+ SMT)”, which reviewers view as fragile and not clearly aligned with mathematical usefulness. Missing are robustness checks (multiple judges, prompt sensitivity), human validation, or a closer analogue to Minimo’s held-out theorem-proving benchmark.
 - Reporting rigor still may be insufficient (kAWp, SsWm). Reviewers explicitly ask for per-iteration values, variance/error bars, and statistical testing. Authors retract “super-linear growth,” but unless the revised paper adds uncertainty quantification, the empirical claims remain hard to trust.
 - Baselines / ablations may remain confounded (JUi9, SsWm). Even if “No usefulness training” is added, concerns persist that multiple changes (library learning, triviality filtering, novelty bias) blur causal attribution. It may remain unclear what portion of the gains comes from the usefulness objective versus engineering stabilizers.
 - Generality and scale (kBd3, kAWp, JUi9). The “trained-from-scratch small model” justification is accepted as a scientific control, but it leaves the central open question, does the approach transfer to stronger provers, larger models, or Lean/Isabelle-like settings. Reviewers wanted either direct comparisons or at least a pilot/transfer experiment. Rebuttal mostly defers to future work.
 - Domain-specific anomaly unresolved (SsWm). Group theory behavior (weaker or reversed trends) still asks for real error analysis beyond “complementary diversity.” Reviewer wanted to know why usefulness scoring appears less effective there and what concrete failure modes look like.

**Reviewer Concerns:**

Here are some of the issues that remained.

 - Underspecified benchmark construction and scoring mechanics (JUi9, SsWm). How the “hard” benchmark set is formed/updated and how subsampling/credit assignment works is still the reproducibility and interpretation risk.
 - Even after rebuttal, the work can read as “Minimo + different selection signal + heuristics.”
 - Weak external validation of “usefulness” (SsWm, kAWp, JUi9). Heavy reliance on LLM-as-judge plus SMT filtering remains fragile without robustness checks or human/held-out task grounding.
 - Insufficient statistical reporting (kAWp, SsWm). Missing or unclear variance, error bars, and significance testing leave the strength of the empirical claims uncertain.
 - Confounded ablations and baselines (JUi9, SsWm). Multiple simultaneous changes make it hard to attribute gains specifically to the usefulness objective rather than added heuristics.
 - Limited generality and scaling evidence (kBd3, kAWp, JUi9). Transfer to stronger provers, larger models, or richer formalisms is still not demonstrated.
 - Unexplained domain failure mode in group theory (SsWm). The anomalous behavior needs deeper error analysis to understand when and why UseForm underperforms or behaves differently.

**Reviewer Scores:**

In the view of the issues above, the score would have probably remained unchanged

---

### Decision · Program_Chairs · 2026-01-26

Reject